# Comparative Transcriptome and Interaction Protein Analysis Reveals the Mechanism of *IbMPK3*-Overexpressing Transgenic Sweet Potato Response to Low-Temperature Stress

**DOI:** 10.3390/genes13071247

**Published:** 2022-07-14

**Authors:** Rong Jin, Tao Yu, Pengyu Guo, Ming Liu, Jiaquan Pan, Peng Zhao, Qiangqiang Zhang, Xiaoya Zhu, Jing Wang, Aijun Zhang, Qinghe Cao, Zhonghou Tang

**Affiliations:** 1Xuzhou Sweet Potato Research Center, Xuzhou Institute of Agricultural Sciences Jiangsu, China/Key Laboratory of Sweet Potato Biology and Genetic Breeding, Ministry of Agriculture/National Agricultural Experimental Station for Soil Quality, Xuzhou 221000, China; jinrong_2012@126.com (R.J.); y875835390@126.com (P.G.); liuming0506@163.com (M.L.); zhaopeng0217@163.com (P.Z.); xhszhang2021@163.com (Q.Z.); zxy15256131797@126.com (X.Z.); wangjing429645671@163.com (J.W.); zhangaijun608@163.com (A.Z.); 19981003@jaas.ac.cn (Q.C.); 2Tube Division, Crop Research Institute, Liaoning Academy of Agricultural Sciences, Shenyang 110000, China; taoyu@cau.edu.cn (T.Y.); pqamy1001@163.com (J.P.)

**Keywords:** *IbMPK3*-overexpressing transgenic sweet potato plants, low-temperature stress, transcriptome analysis, protein interaction

## Abstract

The sweet potato is very sensitive to low temperature. Our previous study revealed that *IbMPK3*-overexpressing transgenic sweet potato (M3) plants showed stronger low-temperature stress tolerance than wild-type plants (WT). However, the mechanism of M3 plants in response to low-temperature stress is unclear. To further analyze how *IbMPK3* mediates low-temperature stress in sweet potato, WT and M3 plants were exposed to low-temperature stress for 2 h and 12 h for RNA-seq analysis, whereas normal conditions were used as a control (CK). In total, 3436 and 8718 differentially expressed genes (DEGs) were identified in WT at 2 h (vs. CK) and 12 h (vs. CK) under low-temperature stress, respectively, whereas 1450 and 9291 DEGs were detected in M3 plants, respectively. Many common and unique DEGs were analyzed in WT and M3 plants. DEGs related to low temperature were involved in Ca^2+^ signaling, MAPK cascades, the reactive oxygen species (ROS) signaling pathway, hormone transduction pathway, encoding transcription factor families (*bHLH*, *NAC*, and *WRKY*), and downstream stress-related genes. Additionally, more upregulated genes were associated with the MAPK pathway in M3 plants during short-term low-temperature stress (CK vs. 2 h), and more upregulated genes were involved in secondary metabolic synthesis in M3 plants than in the WT during the long-time low-temperature stress treatment (CK vs. 12 h), such as fatty acid biosynthesis and elongation, glutathione metabolism, flavonoid biosynthesis, carotenoid biosynthesis, and zeatin biosynthesis. Moreover, the interaction proteins of IbMPK3 related to photosynthesis, or encoding CaM, NAC, and ribosomal proteins, were identified using yeast two-hybrid (Y2H). This study may provide a valuable resource for elucidating the sweet potato low-temperature stress resistance mechanism, as well as data to support molecular-assisted breeding with the *IbMPK3* gene.

## 1. Introduction

The sweet potato (*Ipomoea batatas* (L.) Lam.) is an important crop, which is used as a vegetable, animal feed, an industrial raw material, and a bioenergy source. It is widely cultivated in more than 100 countries in tropical and subtropical regions around the world, due to its rich economic and nutritional value, as well as its excellent adaptability. As the sweet potato can grow well and obtain high yields on barren and marginal land, it is called an “underground granary” and has great potential for promoting agricultural production and regional economic development. However, the sweet potato is very sensitive to low temperature. It usually ceases to grow when the temperature is below to 15 °C. Compared with other environmental factors, such as drought and high temperature, low temperature is more harmful and limits sweet potato growth, development, yield, and geographical distribution. The primary symptoms of low-temperature stress in plants include reduced photosynthetic capacity, the degradation of phospholipids, altered lipid composition, the excess production of ROS, and impaired membrane permeability. These changes disrupt cellular homeostasis and lead to abnormal growth, including chlorotic leaves, moisture loss, and reduced length and weight.

Plants have evolved a series of mechanisms that allow them to adapt to low-temperature stress. Changes in cellular calcium ion (Ca^2+^) concentrations and an increase in ROS are early responses to low-temperature stress and regulate the low-temperature signaling transduction pathway [1,2]. Ca^2+^ signaling plays a key role in the activation of MAP kinase (MAPK) cascades, which are composed of a set of three sequential phosphorylation of MAPK kinases: kinase (MPKKK/MEKK), a MAPK kinase (MAPKK/MKK), and a MAPK (MPK). MAPK cascades respond to various environmental stresses by inducing phosphorylation of diverse downstream effector proteins [3]. The typical MAPK cascade signaling components MEKKK1-MKK2-MPK4/MPK6 in *Arabidopsis* and OsMKK6-OsMPK3 in rice regulate the low-temperature stress response [4,5]. It has been reported that several transcription factors (TFs), including the *AP2/ERF*, *NAC*, *bZIP*, *WRKY,* and *MYB* gene families, are highly enriched under low-temperature stress conditions and are involved in low-temperature stress response [6]. The ICE-CBF-COR response pathway is the classical plant defense mechanism against low-temperature stress [7]. Dehydration-responsive element-binding/C-repeat-binding factor (DREB1/CBF) belongs to the AP2/ERF TFs and plays a leading role in low-temperature stress [8]. *CBFs* can be rapidly induced when plants are exposed to low temperature and activate target cold-regulated genes (*COR*), resulting in enhanced low-temperature stress tolerance in plants [9]. ICE1 is the best *CBF* transcription activator found to date. It can combine with *CBF* promoter to promote *CBF* expression to resist low-temperature stress [10]. Calmodulin (CAMTA), another transcriptional activator of *CBF*, positively regulates the expression of *CBF1* under low-temperature stress [11]. However, MYB15, a member of the R2R3 family, negatively regulates the expression of *CBF* [12]. In addition, there is cross-regulation between low-temperature signaling and the plant hormone transduction pathway [13]. Salicylic acid (SA) mediates the low-temperature-induced grapevine bud dormancy release and effectively alleviates low-temperature inhibition in the growth of winter wheat [14]. Jasmonic acid (JA) positively regulates *CBF* signaling by mediating the interaction of ICE1 in *Arabidopsis* [15]. The JA biosynthetic genes (*LOX1* and *LOX2*) and jasmonate resistant 1 (*JAR1*) are regulated by low-temperature stress, leading to an increase of endogenous JA in *A**donis. annua* under low-temperature treatment [16].

Today, preventing low-temperature stress in sweet potato has received increasing attention. The heterologous expression of low temperature-related genes in sweet potato can improve low-temperature stress tolerance. *SCOF-1*, a zinc lip protein in soybean, is regulated by low temperature. The overexpression of this gene in sweet potato plants can alleviate the damage caused by low-temperature stress, by increasing the photosynthetic rate and oxidoreductase activity [17]. Acid ribosomal P3 (*P3B*), an RNA chaperone, plays a role in phosphorylation and post-transcriptional modification and thus enhances the resistance to various stresses. A transgenic sweet potato with overexpressed *AtP3B* showed stronger resistance to low-temperature stress than non-transgenic plants [18]. In our previous studies, we cloned and over-expressed some homologous genes related to low temperature, to improve the low-temperature tolerance in sweet potato [19,20,21,22,23]. The ICE-CBF-COR pathway involved in the low-temperature transduction pathway has been identified in sweet potato. *IbbHLH79*, an *ICE1*-like gene, can activate the *CBF* pathway [19], and the core region CRT/DRE of *IbCBF3* can bind with the *IbCOR* promoter and induce the expression of IbCORs [20]. *IbbHLH79*-overexpressing sweet potato plants and *IbCBF3*-overexpressing sweet potato plants both displayed enhanced low-temperature stress tolerance.

In our previous study, *IbMPK3* was found to play an important role in responding to low temperature in sweet potato [21]. *IbMPK3* is strongly regulated by low-temperature stress, and *IbMPK3*-overexpressing transgenic sweet potato plants showed stronger low-temperature stress tolerance than wild-type (WT) plants, with higher photosynthesis efficiency, less membrane damage, lower levels of ROS accumulation, and higher levels of enzymatic activities [21]. However, *IbMPK3* could not regulate the CBF-COR pathway, and no significant difference was found in the transcript level of *IbCBF3* between *IbMPK3*-overexpressing transgenic and WT plants. The mechanism of *IbMPK3*-overexpressing transgenic sweet potato response to low-temperature stress is unclear.

RNA-seq has been widely employed in the comparative analysis of plant responses to low temperature [22,23]. Yeast two-hybrid (Y2H) is traditionally used to identify protein interactions, in order to elucidate the functions of target genes. In this study, to further uncover the mechanisms of *IbMPK3*-mediation of the low-temperature signaling pathway and to determine the common and different physiological processes between *IbMPK3*-overexpressing transgenic sweet potato and WT sweet potato during the response to low temperature, differentially expressed genes and pathways between *IbMPK3*-overexpressing transgenic and WT sweet potato plants were analyzed using RNA-seq, and the functions of the proteins that interacted with *IbMPK3* were predicted using Y2H technology. Understanding the mechanism of how *IbMPK3* is involved in the low-temperature signaling pathway and how sweet potato responds to low temperature will provide valuable information for improving the low-temperature tolerance in sweet potato.

## 2. Materials and Methods

### 2.1. Plant Materials and Cold Stress Treatment

IbMPK3-overexpressing transgenic sweet potato plants with increased low-temperature stress tolerance, named M3 plants, were verified in our previous study. The terms WT plants and M3 plants are used in this study to distinguish the non-transgenic sweet potato plants (cv. Xushu29) and *IbMPK3*-overexpressing transgenic sweet potato plants. One-month-old test-tube WT and M3 plants plantlets were cultured under 25 °C and photoperiod (16/8 h light/dark cycle) in a greenhouse. The plants were exposed to low temperature at 4 °C, and the third to fifth leaves were collected after low-temperature stress treatment at 2 h and 12 h, with three biological replicates. After low-temperature stress treatment, the sweet potato plants displayed varying degrees of wilting. The collected samples were immediately frozen in liquid-nitrogen and stored at −80 °C for RNA extraction.

### 2.2. RNA Extraction and cDNA Library Construction

RNA was extracted from WT and M3 plants and monitored on 1% agarose gels for primary detection. The RNA purity (OD260/280 and OD260/230) and integrity were measured using a NanoPhotometer^®^ spectrophotometer (IMPLEN, Westlake Village, CA, USA), respectively. To obtain a global overview of the transcriptome relevant to low-temperature stress in WT and M3 plants, cDNA libraries from WT and M3 plants at different time points (CK, 2 h and 12 h) under low-temperature stress were established. Sequencing libraries were generated using an NEBNext^®^ Ultra™ RNA Library Prep Kit for Illumina^®^ (NEB, Ipswich, MA, USA), and library quality was assessed on the Agilent Bioanalyzer 2100 system (PaloAlto, CA, USA).

### 2.3. Sequencing and Data Processing

The library preparations were sequenced on an Illumina Hiseq 2000 platform, according to the manufacturer’s instructions (Illumina, San Diego, CA, USA), and paired-end reads were generated. After removing adapters and filtering low-quality reads and empty reads from the raw data, clean reads with high quality were obtained and assembled using Trinity software. The Q20, Q30, and GC contents were calculated. Qualified clean reads were mapped to the sweet potato reference genome sequence using Tophat2.0 software.

### 2.4. Identification of Differentially-Expressed Genes (DEGs)

The DEG libraries prepared from samples of WT and M3 plants exposed to low-temperature stress, including WT CK, WT 2 h, WT 12 h, M3 CK, M3 2 h, and M3 12 h, were constructed and sequenced. To estimate gene expression levels, the read count for each gene was obtained from the mapping results and normalized using RPKM (the reads per kilobase per million reads) method. Differential expression analysis of two groups was performed using the DESeq R package (1.10.1) based on the negative binomial distribution model [24]. The false discovery rate (FDR) was calculated, to adjust the threshold of the *p* value to ≤0.05 [25]. Transcripts with a minimum of a 2-fold difference in expression ration ≥ 1 and an FDR ≤ 0.01 were considered to be differentially expressed between the two groups.

### 2.5. Functional Annotation and Gene Ontology (GO) and Kyoto Encyclopedia of Genes and Genomes (KEGG) Enrichment Analysis

GO terms were assigned to each assembled gene transcript using the orgiGO web-based program (http://systemsbiology.cau.edu.cn/agriGOv2) accessed on 11 January 2021 for functional annotation [26], and the threshold for significant GO enrichment of DEGs was set at *p*-value ≤ 0.05. The KOBAS 2.0 web-based program (http://www.genome.jp/kegg) accessed on 11 January 2021 was used to test the statistical enrichment of DEGs [27] in the KEGG pathway with a *q* value ≤ 0.01.

### 2.6. qRT-PCR Validation

Ten randomly selected genes were performed qRT-PCR. One microgram of total RNA was transcribed into cDNA using a TIANScript II RT Kit (TIANGEN, Beijing, China) and qRT-PCR was performed using the OneStep Real-Time System (Applied Biosystem, Carisbad, CA, USA), according to the manufacturer’s instructions. The reference gene actin was used for normalization, and three independent biological replicates were used for each sample. The comparative CT method (2^−^^ΔΔCT^ method) [28] was used to analyze the genes expression levels. The ten randomly selected gene IDs were G26028, G48802, G24647, G19689, G22719, G24812, G16071, G44971, G3829, and MSTRG.52707. The specific primers used are listed in Appendix A.

### 2.7. Yeast Two-Hybrid (Y2H) Analysis

One-month-old test-tube plantlets of M3 plants were harvested after exposure to low temperature at 4 °C for 2 h and 12 h, and used to construct the Y2H library. PCR products with full-length *IbMPK3* were purified and digested with SfiI (NEB, Ipswich, MA, USA), then transformed into DH5α, to generate the pBT3SUC-M3 vector. After testing the autoactivation of the bait IbMPK3, the pBT3SUC-M3 vector was transformed into the yeast library with pPR3N vector, and the transformants were grown on SD/−Leu/−Trp/−His selection medium containing 5 mM 3-amino-1,2,4-triazole (3AT) at 30 °C for 10 days. The combination of pNubG-Fe65 and pTSU2-APP was used as the positive control, and pTSU-APP co-transformed with pPR3N was used as the negative control. Colonies with diameter > 2 mm were selected as primary interacting proteins, and the colonies were diluted with sterile water and grown on SD/−Leu/−Trp/−His/−Ade selection medium containing 40 mg/L x-α-Gal at 30 °C for 3–4 days, to evaluate the activity of the LacZ gene. The sequences obtained from the positive interactions were analyzed using the NCBI database, to identify names and functions.

## 3. Results

### 3.1. RNA-seq Data Analysis

RNA samples used in RNA-seq analysis were isolated from the third to the fifth leaves of WT and M3 plants subjected to low-temperature stress treatment and named WT CK, WT 2 h, WT 12 h, M3 CK, M3 2 h, and M3 12 h. In order to ensure that the RNA-seq results were reliable, three biological replicates were performed for each experimental treatment. A total of 783.59 Mb raw reads for 18 samples were generated through RNA-seq (Appendix A). All of the raw data were deposited into the NCBI Sequence Read Archive (SRA) database (accession numbers SRR15963105, SRR15963106- SRR15963110, and SRR15963118-SRR15963130). After filtering, approximately 775.61 Mb clean reads with a 756 bp average length were generated for the libraries. In total, 75% or more of clean reads were successfully aligned to the sweet potato reference genome. Approximately 42,230–47,665 expressed genes and 8187 novel genes were identified, thus providing a massive amount of data for the subsequent expression profiling analysis.

### 3.2. RNA Sequencing Validation by qRT-PCR

To verify the accuracy of the RNA-seq results, 10 genes were randomly selected from WT and M3 plants to perform qRT-PCR analysis for fluorescent quantitative validation, according to their expression pattern (Appendix A). The qRT-PCR results revealed that the gene expression trends were significantly correlated with those obtained from the RNA-seq data (r^2^ = 0.8869, Figure 1), indicating that the RNA-seq results were reliable.

### 3.3. Identification, Functional Annotation, and Cluster Analysis of DEGs

Principal component analysis (PCA) showed the replicates of each treatment clustered together (Figure 2). The assignment of WT CK, M3 CK, and M3 2 h was similar, suggesting that fewer changes occurred between before and after 2 h for low-temperature stress in M3 plants than in the WT. M3 12 h showed a clear separation from WT 12 h, indicating different molecular mechanisms between the WT and M3 plant response to low-temperature stress.

The DEGs with a *p* value < 0.005 and log_2_ values >1 were identified by pairwise comparisons of transcriptome datasets from WT and M3 plants at the three time point under cold stress (Appendix A). In total, 1193 upregulated and 2243 downregulated DEGs were identified in WT at 2 h (vs. CK), whereas only 753 upregulated and 652 downregulated DEGs were identified in M3 at 2 h (vs. CK). A total of 3516 upregulated and 5202 downregulated genes, and 3732 upregulated and 5559 downregulated DEGs, were detected in the WT and M3 plants at 12 h, respectively, as compared with CK, whereas 3683 upregulated and 3456 downregulated DEGs, and 3342 upregulated DEGs and 3475 downregulated DEGs, were differentially expressed in WT and M3 at 12 h, respectively, as compared with those at 2 h (Figure 3). The number of downregulated genes was greater than that of upregulated genes in both WT and M3 plants. This result indicated that the expression levels of many genes might slow the metabolism of cells, whereas other genes were stimulated, in order to acclimate to low-temperature stress conditions.

To determine the potential functions of the DEGs in response to low-temperature stress, GO enrichment and KEGG classifications were implemented. The DEGs of WT CK vs. 2 h, WT CK vs. 12 h, WT 2 h vs. WT 12 h, M3 CK vs. M3 2 h, M3 CK vs. M3 2 h, and M3 2 h vs. 12 h were assigned to similar enriched GO terms consisting of biological process (including metabolic processes, cellular processes, signal-organism processes, and response to stimulus), cellular components (including cell, cell part, organelle, and membrane), and molecular functions (including catalytic activity, binding, transporter activity, and nucleic acid binding transcription factor activity) (Appendix A; Appendix A). Further analysis showed that in most of the GO categories, there were far more upregulated than downregulated genes between WT CK vs. WT 2 h, whereas there were more upregulated genes than downregulated genes between M3 CK vs. M3 2 h. The distinct regulation of DEGs in WT CK vs. WT 2 h and M3 CK vs. M3 2 h may have resulted in the greater low-temperature stress tolerance in M3 plants compared to WT plants during the initial period of low-temperature conditions. The KEGG enrichment was significantly different between WT and M3 plants (Appendix A; Appendix A). Genes involved in the MAPK signaling pathway and plant–pathogen interactions were highly enriched in M3 plants, while genes involved in flavonoid biosynthesis and metabolic pathways were highly enriched in WT plants. These pathways included the biosynthesis of secondary metabolites and phenylpropanoid biosynthesis, which were commonly highly enriched in WT and M3 plants. The results of the GO and KEGG analysis may provide some clues to understanding the transcriptomic profiles of genes involved in sweet potato response to low-temperature stress.

In addition, a trend analysis of DEGs was performed, and eight expression profiles were clustered. In total, 1946 DEGs, 3118 DEGs, 2284 DEGs, and 999 DEGs, divided into four significant profiles (Profiles 0, 3, 4, and 7, *p* < 0.05), were identified in WT, while 2366 DEGs, 3543 DEGs, 2280 DEGs, and 1375 DEGs were divided into the four profiles in M3 plants (Figure 4). The number of DEGs with continuous positive (Profile 7), persistent negative (Profile 3), and continuous negative (Profile 0) responses to low-temperature stress in M3 were greater than those in WT plants. The number of DEGs with persistent positive (Profile 4) responses to low-temperature stress in M3 plants was similar to that in WT plants. The number of DEGs changed over time in WT and M3 plants, corresponding to their distinct responses to low-temperature stress, suggesting that low-temperature stress resistance in the M3 plants occurred through a highly-complex process.

### 3.4. Functional Analysis of Common and Unique DEGs under Low-Temperature Stress

To determine the common and different mechanisms of low-temperature stress tolerance between WT and M3 plants, the common and unique DEGs involved in persistent positive (Profile 4) and continuous positive (Profile 7) responses in WT and M3 plants were selected for further analysis.

Ca^2+^ signaling is an essential second messenger in the low-temperature signal transduction pathway, and protein kinases are responsible for activating TFs and downstream stress-related genes, to cope with low-temperature stress [29]. Plant hormones perform crosstalk with Ca^2+^ signaling and play important roles in low-temperature stress tolerance [30]. In the current study, among the common DEGs in both WT and M3 plants (Appendix A), 210 DEGs (75 DEGs in Profile 4 and 23 DEGs in Profiles 7) encoding Ca^2+^-binding proteins and protein kinases were identified, including calcium-transporting Ca^2+^ sensor protein calmodulin-like (CML), MAPK cascades (MAPKKK, MAPKK, and MAPK), serinethreonine-protein kinase (STK), leucine-rich repeat recepotor-like kinase (LRR), and CBL-interacting protein kinase (CIPK) (Figure 5A and Figure 6A; Appendix A). A total of 121 DEGs (28 DEGs in Profile 4 and 9 DEGs in Profiles 7) encoding TFs were identified as *WRPK*, *bHLH*, *NAC,* and *ZIP* (Figure 5C and Figure 6C). A total of 114 hormone-related DEGs (28 DEGs in Profile 4 and 16 DEGs in Profiles 7) were found to be associated with metabolic synthesis and the transduction pathways of various hormones, such as ABA, JA, SA, and IAA (Figure 5B and Figure 6B). A total of 80 stress-related DEGs (17 DEGs in Profile 4 and three DEGs in Profiles 7) were identified (Figure 5D and Figure 6D). These DEGs were mainly involved in osmotic regulation, as well as antioxidant and stress response.

Among the DEGs for M3 plants (Appendix A), there were 128 DEGs (25 DEGs in Profile 4 and 47 DEGs in Profiles 7) encoding Ca^2+^-binding proteins and protein kinases (Figure 7A and Figure 8A; Appendix A), 69 DEGs (seven DEGs in Profile 4 and 11 DEGs in Profiles 7) encoding TFs (Figure 7C and Figure 8C), 45 stress-related DEGs (four DEGs in Profile 4 and four DEGs in Profiles 7) (Figure 7D and Figure 8D), and 38 hormone-related DEGs (five DEGs in Profile 4 and eight DEGs in Profiles 7) (Figure 7C and Figure 8C). Additionally, to further determine why the M3 plants showed more tolerance to low-temperature stress than WT, the DEGs that were expressed more rapidly in M3 plants than in WT plants under low-temperature stress were also identified, including 42 DEGs involved in Ca^2+^-binding proteins and protein kinase, 32 DEGs encoding TFs, 31 hormone-related DEGs, and 18 stress-related DEGs (Figure 9; Appendix A). The DEGs in M3 may increase their tolerance to low-temperature stress.

### 3.5. Analysis of MAPK Pathway in WT and M3 Plants during Short-Term Low-Temperature Stress

MAPK is composed of MAPKKK, MAPKK, and MAPK, and is a molecular signal widely found in eukaryotes. It is responsible for the amplification and transmission of intracellular signals through phosphorylation. Many genes are involved in the MAPK pathway and play vital roles in abiotic stress responses, hormone responses, cell division, and growth and development [31,32]. In this study, the MAPK pathway responded different between WT and M3 plants during short-term (CK vs. 2 h) low-temperature stress (Figure 10A; Appendix A). The expression level of *IbMPK3* increased to a higher level in M3 plants than in WT plants, resulting in different expression levels of the downstream genes involved in defense resistance, including *WRKY33*, *WRKY22/29*, ACC synthase (*ACS6*), and chitinase (*CHIB*). There were five genes encoding *WRKY33* (one upregulated and four downregulated), three downregulated genes encoding *ACS6*, and nine downregulated genes encoding *CHIB* in WT plants, while there was one upregulated gene encoding *WRKY33*, one upregulated gene encoding *WRKY22/29*, four upregulated genes encoding *ACS6,* and six upregulated genes encoding *CHIB* in M3 plants. In addition, the expression levels of some other MAPK cascade downstream genes were also regulated in M3 and WT plants. *RbohD* (respiratory burst oxidase homolog; NADPH oxidase), a player in ROS accumulation and *CAT1*, is an H_2_O_2_ cleaner. These key DEGs in the MAPK signaling pathway, combined with the hormone signaling pathway and the ROS pathway, may contribute to the strong low-temperature tolerance in M3 plants.

### 3.6. Analysis of Secondary Metabolic Pathway and ABA-Independent Pathway in WT and M3 Plants during Long-Term Low-Temperature Stress Conditions

The ABA receptor family (PYR/PYL), protein phosphatase 2C (PP2C), and serine/threonine-protein kinase SnRK2 are involved in the ABA-independent pathway response to stresses [33]. In this study, six genes encoding *PYR/PYL* (two upregulated and four downregulated), eight genes encoding *PP2C* (four upregulated and four downregulated), and nine genes encoding *SnRK2* (six upregulated and three downregulated) in WT under low-temperature stress were found, whereas four genes encoding *PYR/PYL* (two upregulated and two downregulated), five genes encoding *PP2C* (four upregulated and one downregulated), and four genes encoding *SnRK2* (two upregulated and two downregulated) in WT under low-temperature stress were identified (Figure 10B; Appendix A). In addition, plants can maintain osmotic homeostasis and increase the content of fatty acids in membrane lipids via the ROS scavenging system, to maintain normal physiological function under low-temperature stress [34]. The accumulation of secondary metabolites can effectively remove the ROS-induced by low-temperature stress [35,36]. In this study, typical secondary metabolic pathways were screened, and it was found that there were more upregulated genes involved in the secondary metabolic pathways in M3 plants than in WT plants (Appendix A). For example, eight genes (three upregulated and five downregulated) were involved in fatty acid biosynthesis in WT, and 11 genes (seven upregulated and four downregulated) were involved in M3 plants; 12 genes (six upregulated and six downregulated) were involved in fatty acid elongation in WT, and 12 genes (four upregulated and eight downregulated) were involved in M3 plants; 68 genes (11 upregulated and 58 downregulated) were involved in glutathione metabolism in WT plants, and 58 genes (24 upregulated and 34 downregulated) were involved in M3 plants; 28 genes (6 upregulated and 22 downregulated) were involved in flavonoid biosynthesis in WT, and 28 genes (12 upregulated and 16 downregulated) were involved in M3 plants; 22 genes (19 upregulated and 6 downregulated) were involved in carotenoid biosynthesis in WT, and 25 genes (19 upregulated and 3 downregulated) were involved in M3 plants; seven genes (2 upregulated and 5 downregulated) were involved in zeatin biosynthesis in WT, and 15 genes (7 upregulated and 8 downregulated) were involved in M3 plants.

### 3.7. IbMPK3 Interact Proteins Analysis

A total of 27 positive clones were tested by growing on SD/−Leu/−Trp medium and SD/−Leu/−Trp/−His/−Ade medium and evaluated with β-galactosidase activity analysis. Except for the No. 6 and No. 21 clones, the clones could grow well on SD/−Leu/−Trp/−His/−Ade SD/-Trp/-Leu/-His/-Ade medium (Figure 11A). Rotation verification was used to acquire true positive clones before sequencing and BLAST analysis (Figure 11B). The No.13 and No.23 clones were unsuccessfully sequenced, whereas the other clones were successfully sequenced (Figure 11C). Five interaction proteins were related to the photosystem, including chlorophyll a-b binding proteins (LHCII, CP24, and CP29), and photosystem I reaction center subunit proteins (PsaD and PsaK); three interaction proteins were related to ribosomal proteins (RPL3, RPS14, and RPS16); and there was one gene of interaction protein encoding ribulose bisphosphate carboxylase small chain SSU1, one gene encoding CaM, one gene encoding TF NAC2, one gene encoding methylene blue sensitivity protein 1 (MBS1), one glycine-rich protein, one metallothionein (MT), one aquaprin TIP1-1, one RHOMBOID-like protein, one mannose/glucose-specific lectin-like protein, and one (S)-Ureidoglycine aminohydrolase enzyme (Table 1).

## 4. Discussion

The mechanism of the response to low-temperature stress in sweet potato is complex and needs to be explored further. In our previous study, M3 plants showed increased tolerance to low temperature compared to WT plants. *IbMPK3* can regulate the low-temperature signaling transduction pathway without activating CBF3-COR module [21]. In this study, to further determine the common physiological processes between WT and M3 plants, as well as the special physiological processes or genes in M3 plants under low-temperature stress, RNA-seq and H2Y technology were used to evaluate the transcriptomes of WT and M3 plants and analyze the IbMPK3 interacting proteins. There were clear transcriptional differences between the WT and M3 plants, and common mechanisms in WT and M3 in response to low temperature.

Ca^2+^ is commonly found in eukaryotic cells and is a major regulation factor mediating plant adaptive responses to adversity stresses. Low-temperature stress induces a rapid change of the Ca^2+^ concentration in cells and activates Ca^2+^ signaling. Ca^2+^ transporter Ca^2+^ATPase (*ACS*) plays a role in maintaining the balance of Ca^2+^ influx and efflux [37]. It was previously reported that the activity of Ca^2+^-ATPase differed in winter wheat and maize under low-temperature stress [38]. Maize Ca^2+^-ATPase became less and less active as chilling proceeded, while Ca^2+^-ATPase could maintain its activity after a chilling treatment in winter wheat. In the present study, four DEGs encoding *ACS* were found in W3 plants, two common ACSs were found in Profile 4, one specific *ACS* was found in Profile 4, and one specific *ACS* was found in Profile 7 (Figure 6, Figure 7 and Figure 8). Ca^2+^ receptor and Ca^2+^-mediated protein kinases, including CaM, CMLs, CDPKs, and CIPKs, are able to decode the information within the different Ca^2+^ oscillations and convert this information into cell functions [39]. Several previous studies have revealed that calcium sensor protein genes are responsible for transmitting and amplifying low-temperature signals and allow plants to adapt to low-temperature stress [40,41,42,43]. In the current study, CaM, CMLs, CDPKs, and CIPKs were also found to participate in low-temperature stress response in sweet potato (Appendix A), suggesting that these genes might be linked to downstream target proteins to adapt to low temperature.

Rebuilding the balance of ROS (H_2_O_2_, HO· and O_2_^-^) is an adaptive physiological and biochemical strategy in plants under environmental stresses. Low-temperature stress can promote H_2_O_2_ production by activating SnRK2-mediated NADPH via ABA signaling [44]. H_2_O_2_ can also be sensed by leucine-rich-repeat receptor kinase (*HPCA1*), and the accumulation of ROS can further activate Ca^2+^ signaling [45]. Under low-temperature stress, most late-response genes encoding proteines have a protective activity and functions, and can increase the resistance to low-temperatures stress through excess ROS scavenging. In this study, *RbohD* and *CAT1* were upregulated during short-term low-temperatrue stress treatment in both WT and M3 plants (Figure 10A), which indicates that crosstalk occurred between the ROS signaling pathway and low-temperature signaling pathway in sweet potato.

Protein phosphorylation–dephosphorylation is a key aspect of abiotic and biotic stress responses in plants. The DEGs encoding protein kinases, including *STK*, *LRR,* and MAPK cascades, were enriched in WT and M3 plants (Appendix A). Previous studies have shown that the MAPKK–MAPK signal transduction network is involved in low-temperature response in many plants [5,32,46,47]. In this study, two genes (G34001 and G10189) were found to encode *IbMPK3*, and the expression level of *IbMPK3* continued to increase in WT and M3 plants under low-temperature stress. Combined with the findings of our previous study, the results indicated that *IbMPK3* played an important role in resistance to low-temperature stress. However, the expression level of *IbMPK3* rose more rapidly and to a higher level in M3 plants than in WT plants. This resulted in MAPK pathway divergence between WT and M3 plants during the short-term low-temperature stress treatment, thus improving the low-temperature tolerance in M3 plants. MAPK cascades, as some of the earliest activated pathways, are involved in multiple defense responses through crosstalk with the signaling of plant defense hormones and the ROS signaling pathway. There were more upregulated *MPK3* downstream genes in M3 plants than in WT plants during short-term temperature stress treatments, including *WRKY33* and *WRKY22*, which are related to pathogen infection [48]; *ACS6*, which is related to ethylene synthesis [49]; and *ChiB*, which is related to defense response (Figure 10A; Appendix A). Some ROS-related genes are also involved in the MAPK cascade pathway and showed different expression patterns in WT and M3 plants, such as *RbohD* and *CAT*. The different numbers and expression of DEGs involved in the MAPK pathway between WT and M3 may give rise to the different levels of low-temperature resistance in these two sweet potatoes.

Various environmental stresses can induce ABA accumulation. ABA-dependent and ABA-independent signaling occur in response to low temperature in plants. *CBF3* plays a pivotal role in ABA-indenpent gene expression under low temperature [50]. However, the expression pattern of *CBF3* showed no significant difference between WT and M3 plants, as verified by the RT-PCR in a previous study [21] and the RNA-seq data in the current study (Appendix A). This indicates that M3 plants adapt to low-temperature stress mainly through the ABA-dependent pathway. *PYR/PYL*, *PP2C,* and *SnRK2* are the key components required for ABA signaling and ABA-mediated responses [33]. *PYL* and *PP2C* negatively regulate the low-temperature stress pathway, whereas *SnRK2* positively regulates the low-temperature stress pathway [51]. Under low-temperature stress, SnRK2.6 (also called OST1) and SIZ1 can inhibit the degradation of ICE1 via phosphorylation and SUMOylation of ICE1, thus improving plant low-temperature stress tolerance [52]. BTF3 can also undergo phosphorylation by OST1 and stabilize *CBFs* via interaction under low-temperature stress conditions [53]. More downregulated genes encoding *PYR/PYL* and *PP2C*, and more upregulated genes encoding *SnKR2.1,* were found in M3 plants compared to WT plants and were involved in the ABA-dependent mediated low-temperature signaling pathway (Figure 10B; Appendix A), which could improve the low-temperature stress tolerance in M3 plants.

Exogenous JA can effectively enhance low temperature tolerance in plants. Jasmonate ZIM-domain protein (*JAZ1/2*) is a key regulator factor in the JA signaling response pathway [54]. The receptor coronatine insensitive 1(*COI1*) senses the JA signaling molecule, binds with substrate JAZ, and then ubiquitinates JAZ [55]. The downstream TFs inhibited by JAZ, including *R2R3-MYB*, *WD-repeat/bHLH/MYB,* and *MYCs*, can be activated to participate in the resistance stress responses regulated by JA [56]. Two genes encoding JAZ were downregulated by the short-term (2 h) low-temperature stress treatment in WT and M3 plants (Appendix A), indicating that endogenous JA may be induced by low temperature. MYCs interact with JAZ to mediate plant root growth and regulate stress-tolerance. The increased transcript level of *MYCs* in WT and M3 plants under long-term (12 h) low-temperature stress conditions (Figure 6) may contribute to adapting to low-temperature stress in both WT and M3 plants.

EIN3/EIL1 (the EIN3 family) play a vital role in the ethylene signaling pathway and also interact with JAZ [57]. These genes regulate plant resistance to pathogens through mediating JA and the ethylene signaling pathway. In *Arabidopsis*, ethylene receptor EIN3 negatively regulates the expression of *CBF* by binding to the EBS motif in *CBF* promoter, thus reducing low-temperature tolerance [58]. The increased expression level of EIN3 was only detected in M3 plants (Figure 7) and may have improved their low-temperature stress tolerance. *ACS* is an important rate-limiting enzyme that can catalyze SAM to ACC. The expression of several *ACS* genes is upregulated by various stresses, resulting in a large amount of ethylene synthesis in plants [59]. Aux/IAA proteins are auxin-sensitive repressors that negatively regulate auxin signaling via repressing the activity of auxin response TFs (*ARFs*) [60]. In *Arabidopsis*, *DREB/CBFs* TFs can positively regulate *IAA5* and *IAA10* in response to abiotic stress [61]. *Aux/IAA14* can regulate the microRNA-mediated low-temperature stress response in Arabidopsis roots. A recessive mutant (slr1; mutation in *Aux/IAA14*) showed sensitivity to low-temperature stress [62]. In the present study, several common and specific genes encoding *ACS* and *Aux/IAA* were induced by low temperature (Figure 5, Figure 6, Figure 7 and Figure 8), suggesting that ethylene and IAA may be involved in the low-temperature signaling pathway in sweet potato plants.

TFs are essential in the response to low-temperature stress. Previous studies have shown that about 10%–20% of *COR* genes are regulated by *CBF* [9]. In *Arabidopsis*, some other cold-induced TFs, including *WRKY33*, *ERF5*, *MYB40*, *HSF1,* and *ZAT10*, have similar functions to *CBF* and activate *CORs* under low-temperature stress [63]. Ethylene-responsive element binding factor *(ERF)/AP2* was observed in response to low-temperature stress [64]. In *Brassica napus*, the conserved Ala37 in the *ERF/AP2* domain is essential for binding with the DRE element of *CBF3* promoter, thus playing a role in low-temperature resistance [65]. Overexpression of *MbERF12* can increase low-temperature tolerance in *Arabidopsis* with ROS scavenging [66]. In the present study, the *ERF008* and *ERF025* involved in Profile 4, and *ERF3*, *ERF4*, *ERF17,* and *ERF109* involved in Profile 7, were upregulated in WT and M3 plants (Figure 5 and Figure 6). This indicated that ethylene may play a positive role in regulating sweet potato low-temperature stress tolerance and make M3 plants more tolerant than WT plants. *WYRK* had the largest numbers of TFs in response to low temperature in M3 plants. *WRKY* genes have been found to be transcriptionally regulated in response to abiotic stress in plants. WRKY22 acts during cryo-stress acclimation in *Arabidopsis* shoot tips via mediating defense-related genes and osmotic stress response-related genes [67]. Overexpression of watermelon *CIWRKY20* in *Arabidopsis* can improve low-temperature tolerance via increased sensitivity to ABA signaling pathway [68]. More *MYRK* genes were found in M3 plants than in WT plants in response to low-temperature stress in the present study, which may have enhanced the tolerance in M3 plants. *ICE1* belongs to the *bHLH* TF family. However, it cannot be activated by low-temperature stress in sweet potato. Other *bHLH* TFs, such as *bHLH92*, *bHLH130,* and *bHLH137* can be regulated by low-temperature stress in sweet potato (Appendix A) and may play roles in sweet potato low-temperature stress tolerance. A total of 175 *IbNACs* have been identified in sweet potato and responded to various stresses, including cold stress [69]. *SINAM1*, a NAC-type tomato TF, could enhance the low-temperature stress tolerance in transgenic tobacco [70]. *CaNAC064* could interact with low-temperature-induced haplo-proteinase proteins and regulated low-temperature stress tolerance in pepper [71]. In addition, many TFs from the *ZIP*, *bHLH*, *ERF,* and *NAC* protein families are known to function in an ABA-dependent manner under cold stress [72]. In this study, the functional enrichment of TFs from the *bHLH*, *ERF*, *MYB*, *MYC,* and *WRKY* families involved in plant hormone signaling pathways and plant–pathogen interaction pathways and IbMPK3 interacting protein NAC revealed the complex regulatory network of low-temperature stress tolerance in sweet potato.

Many functional genes mainly participate in osmotic regulation and increased antioxidant activity and, thus, allow plants to respond to diverse stresses. Secondary metabolites can protect plants from low-temperature stress via the accumulation of soluble sugar and cryoprotective proteins, as well as with the stabilization of cell permeability [73]. More genes involved in secondary metabolic synthesis pathways were upregulated in M3 plants than in WT plants under low temperature (Appendix A), which may be an important factor protecting W3 plants from low-temperature stress. Additionally, this study screened common and specific stress-related DEGs and found that heat-shock transcription factor (*HSF*) and heat-shock protein (*HSP*) were enriched in common DEGs between WT and M3 plants (Profile 4), and in specific DEGs in M3 plants (Profile 4 and Profile 7) (Figure 5, Figure 6, Figure 7 and Figure 8). There were remarkable differences in the transcriptional regulation of *HSF* and *HSP*, indicating that they have undergone considerable functional diversification. Previous studies used RNA-seq analysis to reveal that *HSF* and *HSP* were positively regulated by low-temperature stress in rice [74], rubber tree [75], and eggplant [76]; these findings were similar to the results obtained in this study. The strong expression of *HSF* and HSP under low-temperature stress indicates that *HSF* and *HSP* may play important roles in the low-temperature signaling pathway.

In our previous study, M3 plants showed a better low-temperature stress tolerance than WT, with a higher photosynthetic capacity. Five IbMPK3 interaction proteins related to photosystem, LHCII, CP24,CP29, PsaD, and PsaK, were found in the current study, which may indicate that these genes play vital roles in M3 plants in alleviating the reduction of photosynthetic capacity induced by low temperature. NAC TFs (NAM, ATAF1/2, and CUC2) are regulators of low-temperature stress in several plants, and camodulin (CaM) protein, as a primary Ca^2+^ sensor, controls diverse cellular functions and adaption to environmental stresses. In this study, one gene encoding NAC and one gene encoding CaM were identified, and interacted with IbMPK3. The results further illustrated that CaM and NAC are involved in regulating low-temperature stress in sweet potato. Ribosomal proteins AtP3B, acting as both proteins and RNA chaperones, increased with low temperature in transgenic Arabidopsis and sweet potato. TCD11 encodes the ribosomal small subunit protein S6 that is essential for chloroplast development at low temperature. Three ribosomal proteins (RPL3, RPS14, and RPS16) interacted with IbMPK3 and may be helpful for resistance to low-temperature stress in sweet potato (Table 1). The interaction protein analysis encouraged us to further explore the molecular functions of these genes involved in low-temperature stress.

## 5. Conclusions

In this study, we obtained an overview of the transcriptome of WT and M3 plants under low temperature and analyzed the IbMPK3 interaction proteins. In total, 3436 DEGs and 8718 DEGs were identified in WT at 2 h (vs. CK) and 12 h (vs. CK) under low-temperature stress, respectively, whereas 1450 and 9291 DEGs were detected in M3 plants at 2 h (vs. CK) and 12 h (vs. CK), respectively. Numerous common and unique DEGs were observed in WT and M3 plants. The upregulated low-temperature stress-responsive DEGs involved in Profile 4 and Profile 7 were related to Ca^2+^ signaling, MAPK cascades, the ROS signaling pathway, hormone transduction pathway, and TF families (including *bHLH*, *NAC,* and *WRKY*). The MAPK pathway and secondary metabolic pathway played important roles in low-temperature stress tolerance in M3 plants (Figure 12). The interaction proteins related to photosynthetic or encoding CaM, NAC, and ribosomal proteins may also regulate the low-temperature stress tolerance in sweet potato. These findings will help to elucidate the mechanism of the *IbMPK3*-mediated low-temperature signaling pathway and provide gene resources for molecular-assisted breeding for improving the low-temperature stress tolerance in sweet potato.

## Figures and Tables

**Figure 1 genes-13-01247-f001:**
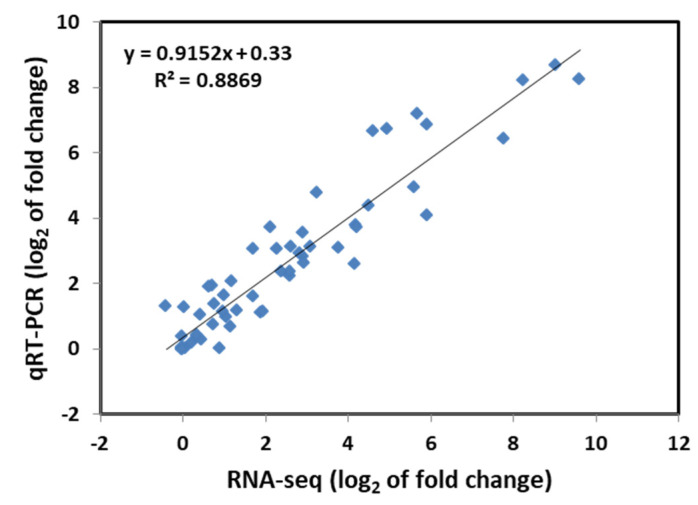
Correlation analysis of the relative expression of randomly selected genes obtained from qRT-PCR and RNA-seq data. The log_2_ ratio values of relative expression are between non-low-temperature stress-treated and low-temperature-treated samples.

**Figure 2 genes-13-01247-f002:**
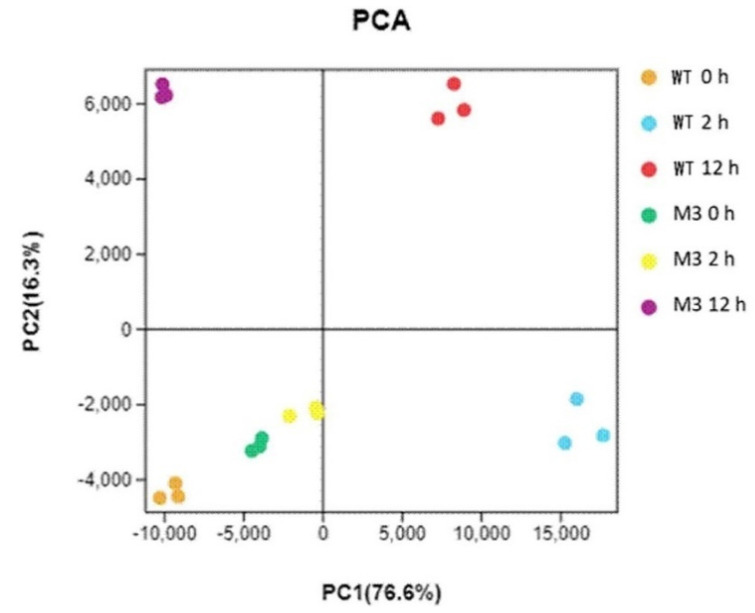
Principal component analysis (PCA) for all RNA-seq samples. WT CK, WT 2 h, and WT 12 h represents WT samples before low-temperature stress treatment and obtained at 2 h and 12 h after low-temperature stress treatment, respectively. M3 CK, M3 2 h, and M3 12 h represent M3 plant samples before low-temperature stress treatment and obtained at 2 h and 12 h after low-temperature stress treatment, respectively.

**Figure 3 genes-13-01247-f003:**
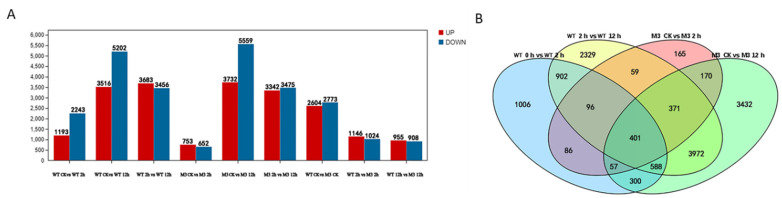
Profiles of gene expression in WT and M3 plants exposed to low-temperature stress for 2 h and 12 h. (**A**) The total number of upregulated and downregulated genes in WT and M3 plants under low-temperature stress treatment. (**B**) Venn diagram describing overlaps among DEGs in WT and M3 plants under low-temperature stress treatment.

**Figure 4 genes-13-01247-f004:**
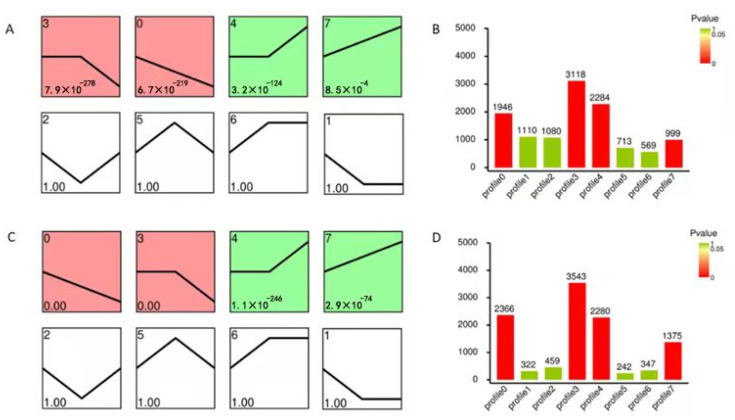
Expression profiles of DEGs involved in the eight clusters. Patterns of gene expression across three time points (CK, 2 h, and 12 h) under low-temperature stress treatment in WT (**A**) and M3 plants (**C**). Profile 0 and Profile 3 colored with red in (**A**,**C**) indicated up-regulated expression tendency, while Profile 4 and Profile 7 colored with green indicated down-regulated expression tendency (*p* ≤ 0.05). The Profile 1, 2, 5, 6 in (**A**,**C**) filled with white color indicated *p* value greater than 0.05. In each profile, the black line represents the expression tendency. The top left-hand corner indicates the number of profiles, while the lower left-hand corner contains the *p*-values of the profile. The number of genes belonging to each cluster is shown in the WT (**B**) and in M3 plants (**D**).

**Figure 5 genes-13-01247-f005:**
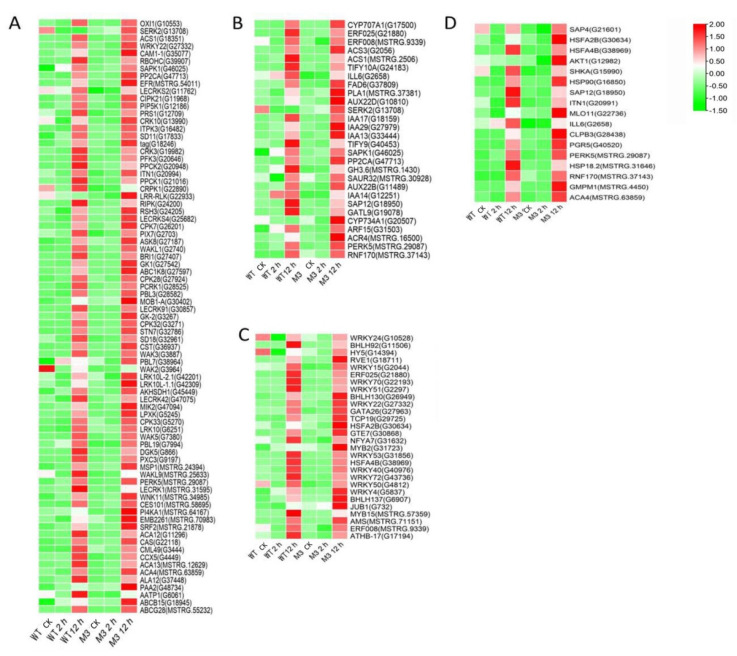
Heatmap of common DEGs involved in Profile 4 in both WT and M3 plants. (**A**) Protein kinase and Ca^2+^ signal-related genes, (**B**) hormone-related genes, (**C**) transcription factors, and (**D**) stress-related genes were identified. The sample and treatments are displayed at the bottom of each column. Relative expression levels are shown by a colored gradient, from low (green) to high (red). Scale bars represent the log_2_ transformations of the RPKM values.

**Figure 6 genes-13-01247-f006:**
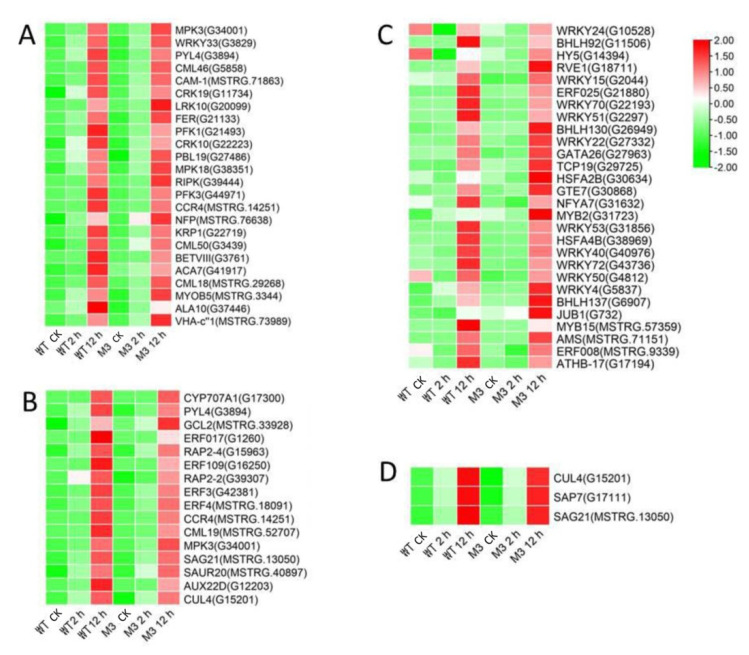
Heatmap of common DEGs involved in Profile 7 in both WT and M3 plants. (**A**) Protein kinase and Ca^2+^ signal-related genes, (**B**) hormone-related genes, (**C**) transcription factors, and (**D**) stress-related genes were identified. The sample and treatments are displayed at the bottom of each column. Relative expression levels are shown by a colored gradient, from low (green) to high (red). Scale bars represent the log_2_ transformations of the RPKM values.

**Figure 7 genes-13-01247-f007:**
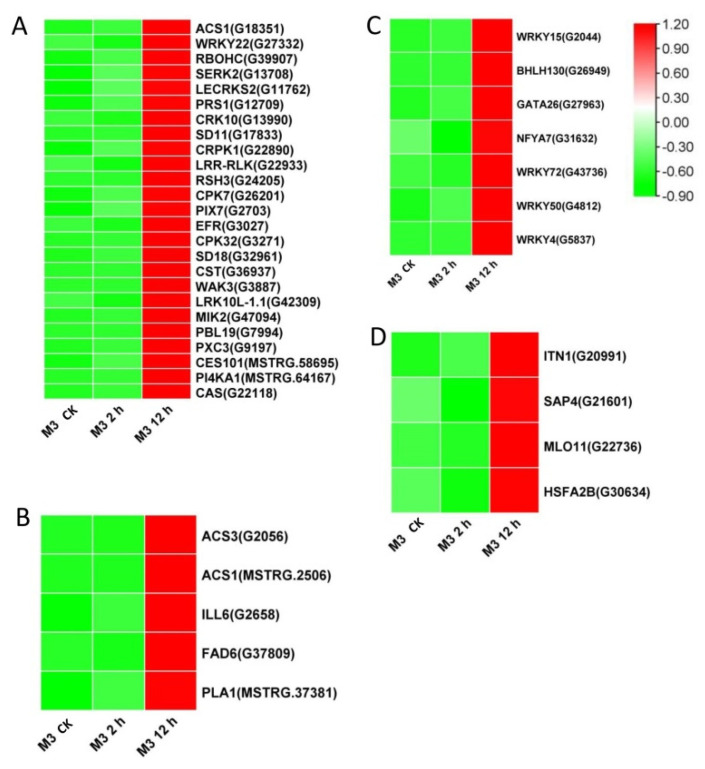
Heatmap of DEGs unique to M3 plants involved in Profile 4 in M3 plants. (**A**) Protein kinase and Ca^2+^ signal-related genes, (**B**) hormone-related genes, (**C**) transcription factors, and (**D**) stress-related genes were identified. The sample and treatments are displayed at the bottom of each column. Relative expression levels are shown by a colored gradient, from low (green) to high (red). Scale bars represent the log_2_ transformations of the RPKM values.

**Figure 8 genes-13-01247-f008:**
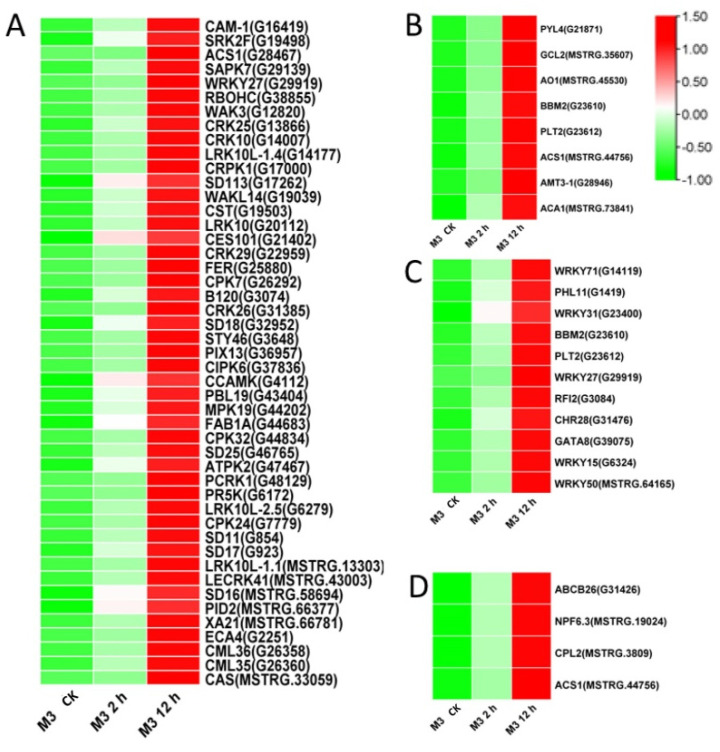
Heatmap of DEGs unique to M3 plants involved in Profile 7 in M3 plants. (**A**) Protein kinase and Ca^2+^ signal-related genes, (**B**) hormone-related genes, (**C**) transcription factors, and (**D**) stress-related genes were identified. The sample and treatments are displayed at the bottom of each column. Relative expression levels are shown by a colored gradient, from low (green) to high (red). Scale bars represent the log_2_ transformations of the RPKM values.

**Figure 9 genes-13-01247-f009:**
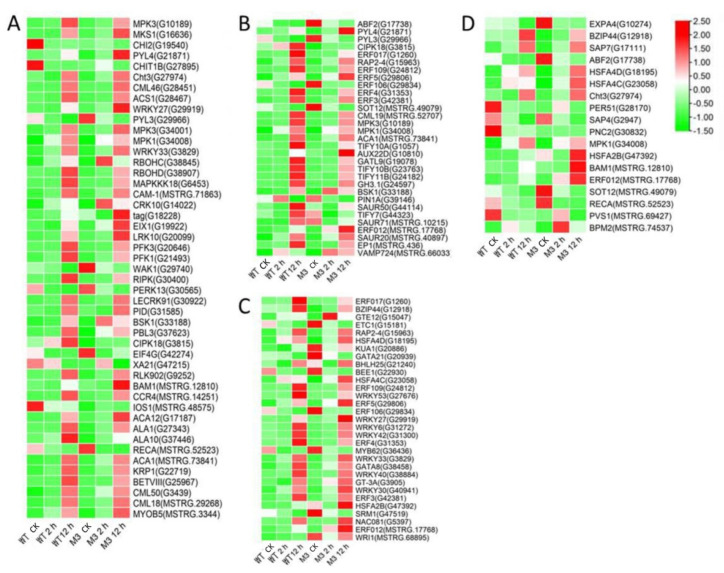
Heatmap of DEGs expressed faster in M3 plants than in WT plants. (**A**) Protein kinase and Ca^2+^ signal-related genes, (**B**) hormone-related genes, (**C**) transcription factors, and (**D**) stress-related genes were identified. The sample and treatments are displayed at the bottom of each column. Relative expression levels are shown by a colored gradient, from low (green) to high (red). Scale bars represent the log_2_ transformations of the RPKM values.

**Figure 10 genes-13-01247-f010:**
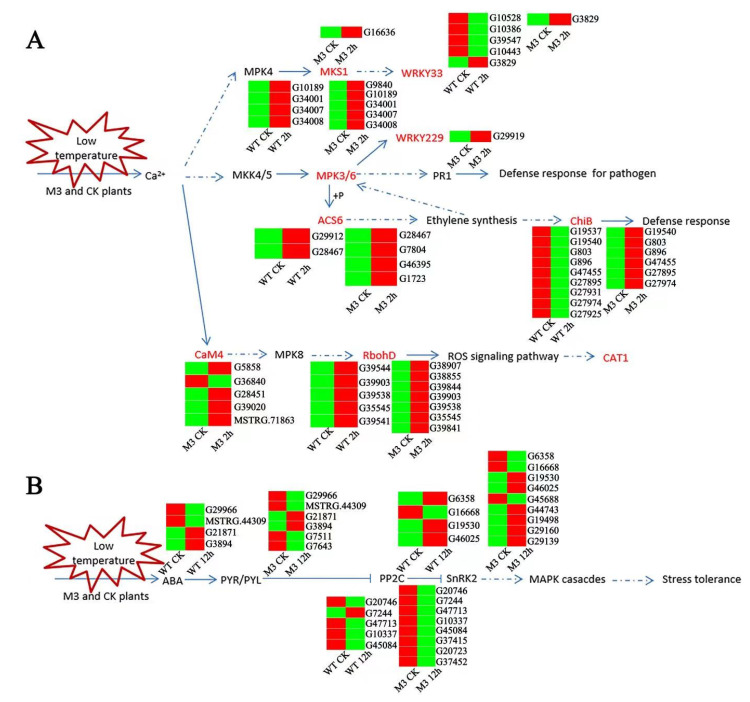
Hypothetical schematic model showing the MAPK pathway and ABA-independent pathway in WT and M3 plants. (**A**) Hypothetical schematic model of MAPK pathway during short-term (CK vs. 2 h) low-temperature stress. (**B**) Hypothetical schematic model of ABA-independent pathway during long-term (CK vs. 12 h) low-temperature stress. An MAPK pathway and ABA-independent pathway model was developed based on the KO enrichment pathway Map04016 and Map04075, respectively. Unique DEGs are shown by heatmap. The red and green color indicated up-regulated and down-regulated expression pattern of genes in heatmap. Blue lines with arrows denote certain regulation, and blue dotted lines with arrows denote uncertain regulation.

**Figure 11 genes-13-01247-f011:**
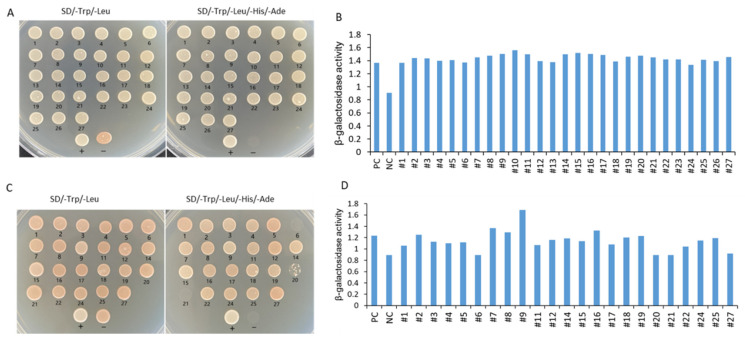
Identification of IbMPK3 interaction proteins using Y2H screening. (**A**) A total of 27 positive clones were screened and found to grow on SD/-Trp/-Leu medium and SD/-Trp/-Leu/-His/-Ade medium. “+” represents the positive control of the yeast two-hybrid, the combination of pNubG-Fe65 and pTSU2-APP. “−” shows the negative control of yeast two-hybrid, the pTSU-APP was co-transformed with pPR3N. (**B**) Semi-quantitative analysis of β-galactosidase activity in 27 initially positive clones. PC indicates positive control, and NC indicates negative control. (**C**) Rotational verification was used to acquire true positive clones after sequencing and BLAST analysis. The clones with number 13 and number 23 could not be sequenced, and the clones with number 6 and number 21 could not be grown on SD/-Trp/-Leu/-His/-Ade medium. (**D**) Semi-quantitative analysis of β-galactosidase activity in successfully sequenced positive clones.

**Figure 12 genes-13-01247-f012:**
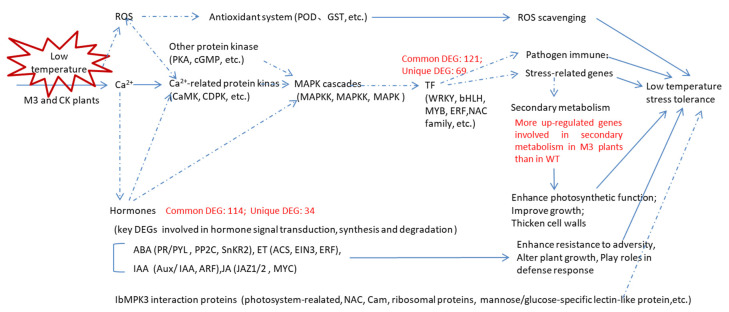
Hypothetical schematic model showing the mechanism in WT and M3 plants against low-temperature stress. This model was developed by combining the results of the RNA-seq analysis with previously described schemes of plant abiotic stress response networks. Blue lines with arrows denote a certain regulation; and blue dotted lines with arrows denote an uncertain regulation. The various crosstalk and feedback of multiple pathways, such as the hormone signaling pathway, ROS signaling pathway, and secondary metabolic pathway, as well as the genes encoding TF families and stress-related genes are assumed to respond to low-temperature stress.

**Table 1 genes-13-01247-t001:** Sequence and Blast of IbMPK3 interacting proteins.

Number	NCBI Accession	NCBI Description
1	XM_031267007.1	PREDICTED: *Ipomoea triloba* calmodulin-7 (LOC116025682), mRNA (84–633)
2	XM_031270274.1	PREDICTED: *Ipomoea triloba* 40S ribosomal protein S16-like (LOC116028540), mRNA (193–663)
3	XM_031274641.1	PREDICTED: *Ipomoea triloba* chlorophyll a-b binding protein of LHCII type 1-like (LOC116032192), mRNA (31–769)
4	XM_031244518.1	PREDICTED: *Ipomoea triloba* photosystem I reaction center subunit IV B, chloroplastic (LOC116004457), mRNA (1–540)
10		PREDICTED: *Ipomoea triloba* photosystem I reaction center subunit IV B, chloroplastic (LOC116004457), mRNA (1–586)
5	XM_031252462.1	PREDICTED: *Ipomoea triloba* ribulose bisphosphate carboxylase small chain SSU11A, chloroplastic-like (LOC116012798), mRNA (274–786)
7	XM_031272632.1	PREDICTED: *Ipomoea triloba* peptidyl-prolyl cis-trans isomerase-like (LOC116030396), mRNA (29–776)
8	XM_031276471.1	PREDICTED: *Ipomoea triloba* 60S ribosomal protein L3 (LOC116033703), mRNA (652–1447)
9	XM_031272345.1	PREDICTED: *Ipomoea triloba* (S)-ureidoglycine aminohydrolase (LOC116030182), mRNA (152–1218)
11	XM_031239861.1	PREDICTED: *Ipomoea triloba* NAC domain-containing protein 2-like (LOC115999908), mRNA (61–803)
26		PREDICTED: *Ipomoea triloba* NAC domain-containing protein 2-like (LOC115999908), mRNA (61–1047)
12	XM_031258537.1	PREDICTED: *Ipomoea triloba* translationally-controlled tumor protein homolog (LOC116017878), mRNA (166–709)
14	XM_031241907.1	PREDICTED: *Ipomoea triloba* mannose/glucose-specific lectin-like (LOC116001939), mRNA (363–672)
15	XM_019333621.1	PREDICTED: *Ipomoea nil* RHOMBOID-like protein 2 (LOC109183524), mRNA (
16	XM_031275192.1	PREDICTED: *Ipomoea triloba* chlorophyll a-b binding protein CP24 10A, chloroplastic (LOC116032562), mRNA (62–803)
17	XM_031246527.1	PREDICTED: *Ipomoea triloba* 40S ribosomal protein S14 (LOC116006230), mRNA (78–754)
18	XM_031246505.1	PREDICTED: *Ipomoea triloba* protein METHYLENE BLUE SENSITIVITY 1-like (LOC116006203), mRNA (75–680)
19	XM_031272347.1	PREDICTED: *Ipomoea triloba* aquaporin TIP1-1-like (LOC116030185), mRNA (54–1070)
22	XM_031272599.1	PREDICTED: *Ipomoea triloba* chlorophyll a-b binding protein CP29.1, chloroplastic-like (LOC116030369), mRNA (617–1076)
24	XM_031257562.1	PREDICTED: *Ipomoea triloba* glycine-rich protein-like (LOC116017048), mRNA (239–526)
25	MG001450.1	*I*.*batatas* metallothionein mRNA, complete cds (2–563)
27	XM_031255147.1	PREDICTED: *Ipomoea triloba* photosystem I reaction center subunit psaK, chloroplastic (LOC116015131), mRNA (90–597)

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
