# Peer review of "Comparative Transcriptome and Interaction Protein Analysis Reveals the Mechanism of IbMPK3-Overexpressing Transgenic Sweet Potato Response to Low-Temperature Stress"

_genes, 2022, doi:10.3390/genes13071247_

Round 1

Reviewer 1 Report

The manuscript presents a study of the response of transgenic sweet potato to low temperature stress by comparative transcriptome. Their result suggest that the mechanism of IbMPK3-overexpression mediated low temperature stress. This is a quite good study in the plant field. However, the manuscript currently needs some minor revision, which precludes the acceptance of the current version at Genes. Presenting more details on treatment condition (light/ dark cycle during treatment, stage of plant growth when the treatment applied…), RNA-seq (read length) is therefore suggested to improve the manuscript.

Author Response

Thanks very much for taking your time to review this manuscript. I really appreciated all your comments and suggestions! The suggestions have enabled us to improve our work. We added some details and modified some unclear expression as your suggestion, and marked them in red in the modified manuscript. Please find my itemized responses in below and my corrections in the re-submitted files. Thanks again.

We added some detail on plants treatment condition and RNA-seq in the manuscript, such as 16/8 h light/dark cycle, the stage of plant growth when the treatment applied.

The modified or added sentence as follows:

1) One-month-old test-tube WT and M3 plants plantlets were cultured under 25°C and photoperiod (16/8 h light/dark cycle) in a greenhouse. (M&M 2.1)

2) After cold stress treatment, sweetpotato plants displayed varying degrees of wilting. (M&M 2.1)

3) After filtering, approximately 775.61 Mb clean reads with 756bp average length were generated for the libraries. (Results 3.1)

Reviewer 2 Report

The manuscript titled "Comparative transcriptome and interaction proteins analysis 2 reveals the mechanism of IbMPK3-overexpressing transgenic 3 sweetpotato response to low temperature stress" represents novel knowledge on the mechanisms that sweetpotato employ to overcome cold stress. The study is scientifically sound, the goal is well presented, methods are properly used, and discussion is sufficiently wide to deal with most represented issues in this scientific field. However, although well explained in the text, some results might be better visually presented.

More precisely:

Figure 3 (both A and B) contains very tiny labels that are fully unreadable. The figure should be enlarged and resolution improved.

Scale bar in Figs 7, 8, and 9 is barely visible. Can the authors increase the font and resolution?

Figure 10 is of very low resolution and it must be improved in this sense.

Figure 11 contains letters of too small font size in B and D. They should be enlarged.

One more major point is that the control treatment without low temperature stress (0 h) is presented as a low temperature treatment. Please rectify the manuscript wherever it is mentioned towards presenting this accession as a control but not as a treatment.

Please cite "our previous study" in L98 and L122.

Please introduce the Latin name and species authority for sweetpotato in L37.

Please list 10 randomly selected genes in L200.

Many typographical and syntax issues are seen. Please correct:

L25: "duuring"

L52: "a series of mechanism"

L56: add "of" after "phosphorylation"

L86: "by the  increasing"

L87: "plays a function" into "has a function" of "plays a role"

L89: "with overexpressed of"

L90: "shown" into "showed" or "has shown"

L132: delete "respectively". It is not applicable here.

L212: "fewere"

L246: "analyis"

L255: "plantys"

L275 and 276: please decide upon "Profile" or "profile".

Do not start a sentence with a digit in L292, L408, L587, and L598.

L298: "as well as and"

L312: write "Among the DEGs unique for M3 plants" instead. Please implement the same structure in L325 and L332.

L318: "Additional" into "Additionally"

L348-349: "responded different between in"

L359-360: The sentence lacks a verb.

L369: "are shown" instead of "were showed". Do not capitalize "heatmap". Do not use semicolon in front of "and". In L622 too.

L402-405: Something strange has happened with font size here.

L419: "were screen"

L421: Do not capitalize "shows".

L425: "were failed to sequenced"

Table 1: Please italicize "Ipomoea triloba", "Ipomoea nil", and "Ipomoea batatas" throughout.

L546: "indicates" instead of "indicated"

L569: "increased"

Many double spaces are noted as well as lack of spaces.

Author Response

Thanks very much for taking your time to review this manuscript. I really appreciated all your comments and suggestions! The suggestions have enabled us to improve our work. We improved resolution and enlarged the font of figures as your suggestion and we corrected wrong presentation, spelling and grammatical expression, and marked them in red color in the modified manuscript. Please find my itemized responses in below and my corrections in the re-submitted files. Thanks again.

Q1. Figure 3 (both A and B) contains very tiny labels that are fully unreadable. The figure should be enlarged and resolution improved.

A1: The resolution of Figure 3 has been improved. Please find the new figure in the modified manuscript.

Q2. Scale bar in Figs 7, 8, and 9 is barely visible. Can the authors increase the font and resolution?

A2: Scale bar in Figures 7, 8, and 9 has been enlarged and improved. Please find the new figure in the modified manuscript.

Q3. Figure 10 is of very low resolution and it must be improved in this sense.

A3: The resolution of Figure 10 has been improved. Please find the new figure in the modified manuscript.

Q4. Figure 11 contains letters of too small font size in B and D. They should be enlarged.

A4: The font size of Figure 11 B and C has been enlarged. Please find the new figure in the modified manuscript.

Q5. One more major point is that the control treatment without low temperature stress (0 h) is presented as a low temperature treatment. Please rectify the manuscript wherever it is mentioned towards presenting this accession as a control but not as a treatment.

A5: We have rectified the wrong presentation. We changed 0 h into CK in the modified manuscript wherever it is mentioned, including figures and tables.

Q6. Please cite "our previous study" in L98 and L122.

A6: We have added the references of "our previous study" in L98 and L122.

Q7. Please introduce the Latin name and species authority for sweetpotato in L37.

A7: We have added the Latin name and species authority for sweetpotato in L37.

Q8. Please list 10 randomly selected genes in L200.

A8: The ten randomly selected genes ID for qRT-PCR are G26028, G48802, G24647, G19689, G22719, G24812, G16071, G44971, G3829 and MSTRG.52707. We have listed them in L206-207 (2.6 qRT-PCR validation).

Q9. Many typographical and syntax issues are seen. Please correct:

L25: "duuring"

L52: "a series of mechanism"

L56: add "of" after "phosphorylation"

L86: "by the  increasing"

L87: "plays a function" into "has a function" of "plays a role"

L89: "with overexpressed of"

L90: "shown" into "showed" or "has shown"

L132: delete "respectively". It is not applicable here.

L212: "fewere"

L246: "analyis"

L255: "plantys"

L275 and 276: please decide upon "Profile" or "profile".

Do not start a sentence with a digit in L292, L408, L587, and L598.

L298: "as well as and"

L312: write "Among the DEGs unique for M3 plants" instead. Please implement the same structure in L325 and L332.

L318: "Additional" into "Additionally"

L348-349: "responded different between in"

L359-360: The sentence lacks a verb.

L369: "are shown" instead of "were showed". Do not capitalize "heatmap". Do not use semicolon in front of "and". In L622 too.

L402-405: Something strange has happened with font size here.

L419: "were screen"

L421: Do not capitalize "shows".

L425: "were failed to sequenced"

Table 1: Please italicize "Ipomoea triloba", "Ipomoea nil", and "Ipomoea batatas" throughout.

L546: "indicates" instead of "indicated"

L569: "increased"

Many double spaces are noted as well as lack of spaces.

A9: We have modified the typographical and syntax issues in the manuscript and marked them in red color. Please check them.

Thanks very much for taking your time to review this manuscript. I really appreciated all your comments and suggestions! The suggestions have enabled us to improve our work. We improved resolution and enlarged the font of figures as your suggestion and we corrected wrong presentation, spelling and grammatical expression, and marked them in red color in the modified manuscript. Please find my itemized responses in below and my corrections in the re-submitted files. Thanks again.

Q1. Figure 3 (both A and B) contains very tiny labels that are fully unreadable. The figure should be enlarged and resolution improved.

A1: The resolution of Figure 3 has been improved. Please find the new figure in the modified manuscript.

Q2. Scale bar in Figs 7, 8, and 9 is barely visible. Can the authors increase the font and resolution?

A2: Scale bar in Figures 7, 8, and 9 has been enlarged and improved. Please find the new figure in the modified manuscript.

Q3. Figure 10 is of very low resolution and it must be improved in this sense.

A3: The resolution of Figure 10 has been improved. Please find the new figure in the modified manuscript.

Q4. Figure 11 contains letters of too small font size in B and D. They should be enlarged.

A4: The font size of Figure 11 B and C has been enlarged. Please find the new figure in the modified manuscript.

Q5. One more major point is that the control treatment without low temperature stress (0 h) is presented as a low temperature treatment. Please rectify the manuscript wherever it is mentioned towards presenting this accession as a control but not as a treatment.

A5: We have rectified the wrong presentation. We changed 0 h into CK in the modified manuscript wherever it is mentioned, including figures and tables.

Q6. Please cite "our previous study" in L98 and L122.

A6: We have added the references of "our previous study" in L98 and L122.

Q7. Please introduce the Latin name and species authority for sweetpotato in L37.

A7: We have added the Latin name and species authority for sweetpotato in L37.

Q8. Please list 10 randomly selected genes in L200.

A8: The ten randomly selected genes ID for qRT-PCR are G26028, G48802, G24647, G19689, G22719, G24812, G16071, G44971, G3829 and MSTRG.52707. We have listed them in L206-207 (2.6 qRT-PCR validation).

Q9. Many typographical and syntax issues are seen. Please correct:

L25: "duuring"

L52: "a series of mechanism"

L56: add "of" after "phosphorylation"

L86: "by the  increasing"

L87: "plays a function" into "has a function" of "plays a role"

L89: "with overexpressed of"

L90: "shown" into "showed" or "has shown"

L132: delete "respectively". It is not applicable here.

L212: "fewere"

L246: "analyis"

L255: "plantys"

L275 and 276: please decide upon "Profile" or "profile".

Do not start a sentence with a digit in L292, L408, L587, and L598.

L298: "as well as and"

L312: write "Among the DEGs unique for M3 plants" instead. Please implement the same structure in L325 and L332.

L318: "Additional" into "Additionally"

L348-349: "responded different between in"

L359-360: The sentence lacks a verb.

L369: "are shown" instead of "were showed". Do not capitalize "heatmap". Do not use semicolon in front of "and". In L622 too.

L402-405: Something strange has happened with font size here.

L419: "were screen"

L421: Do not capitalize "shows".

L425: "were failed to sequenced"

Table 1: Please italicize "Ipomoea triloba", "Ipomoea nil", and "Ipomoea batatas" throughout.

L546: "indicates" instead of "indicated"

L569: "increased"

Many double spaces are noted as well as lack of spaces.

A9: We have modified the typographical and syntax issues in the manuscript and marked them in red color. Please check them.
